# Synthesis of Highly Potent Anti-Inflammatory Compounds (ROS Inhibitors) from Isonicotinic Acid

**DOI:** 10.3390/molecules26051272

**Published:** 2021-02-26

**Authors:** Sana Yaqoob, Nourina Nasim, Rahila Khanam, Yan Wang, Almas Jabeen, Urooj Qureshi, Zaheer Ul-Haq, Hesham R. El-Seedi, Zi-Hua Jiang, Farooq-Ahmad Khan

**Affiliations:** 1Third World Center for Science and Technology, H.E.J. Research Institute of Chemistry, International Center for Chemical and Biological Sciences, University of Karachi, Karachi 75270, Pakistan; sana.yaqoob@iccs.edu (S.Y.); nourina@iccs.edu (N.N.); rahila.khanam@iccs.edu (R.K.); yan.wang@iccs.edu (Y.W.); urooj.qureshi@iccs.edu (U.Q.); zaheer.qasmi@iccs.edu (Z.U.-H.); 2Dr. Panjwani Center for Molecular Medicine and Drug Research, International Center for Chemical and Biological Sciences, University of Karachi, Karachi 75270, Pakistan; almas@iccs.edu; 3International Research Center for Food Nutrition and Safety, Jiangsu University, Zhenjiang 212013, China; 4Department of Molecular Biosciences, The Wenner-Gren Institute, Stockholm University, S-106 91 Stockholm, Sweden; 5Department of Chemistry, Faculty of Science, Menoufia University, Menoufia 32511, Egypt; 6Department of Chemistry, Lakehead University, 955 Oliver Road, Thunder Bay, ON P7B 5E1, Canada; zjiang@lakeheadu.ca

**Keywords:** anti-inflammatory, ROS inhibitors, isonicotinic acid, pyridine carboxylic acid, molecular docking studies

## Abstract

In search of anti-inflammatory compounds, novel scaffolds containing isonicotinoyl motif were synthesized via an efficient strategy. The compounds were screened for their in vitro anti-inflammatory activity. Remarkably high activities were observed for isonicotinates **5**–**6** and **8a**–**8b**. The compound **5** exhibits an exceptional IC_50_ value (1.42 ± 0.1 µg/mL) with 95.9% inhibition at 25 µg/mL, which is eight folds better than the standard drug ibuprofen (11.2 ± 1.9 µg/mL). To gain an insight into the mode of action of anti-inflammatory compounds, molecular docking studies were also performed. Decisively, further development and fine tuning of these isonicotinates based scaffolds for the treatment of various aberrations is still a wide-open field of research.

## 1. Introduction

Inflammation is a complex defense-related response caused by invading microbes or physical injuries [1]. Reactive-oxygen species (ROS) are the key signaling molecules, which are associated with the progression of inflammatory response cycle [2]. They are broadly described as partially reduced oxygen-containing metabolites with strong oxidising capabilities [3]. These molecules are highly reactive to various intracellular components due to the presence of unstable oxygen atom in them [4]. Recent studies explicitly demonstrate that ROS instigates several signaling cascades within immune cells, which ultimately leads to the activation of innate and acquired immunity. Thus, the enhanced production of ROS plays an important role for the development and progression of inflammatory disorders [3]. Therefore, the suppression of ROS overproduction is a promising approach for the prophylaxis and treatment of various chronic inflammation-related aberrations [5]. Lately, many isonicotinic acid derived compounds are being patented due to their astounding inhibitory activities against certain enzymes, such as myeloperoxidase (MPO), [6] urease, acetylcholinesterase, [6] cyclooxygenase-2 (COX-2), [7] Bcr-Abl tyrosine kinase [8], and histone demethylase [9,10]. In the market, Isoniazid (anti-tuberculosis), Enisamium (antiviral), Ethionamide (anti-tuberculosis), and Dexamethasone isonicotinate (anti-inflammatory and anti-allergic) exemplify isonicotinic acid-derived drugs (Figure 1) [11].

Comprehensive and vivid literature is available that highlight the therapeutic value of this pyridine derivative [10,12,13,14,15,16,17,18]. Notably, Gilani et al. showed that isonicotinic acid-derived 1,3,4-oxadiazoles exhibited an interesting profile of anti-inflammatory activity, which was superior to the standard drug Naproxen [19]. The close congener, 2,6-disubstituted isonicotinic acid hydrazides, also demonstrated an enormous anti-inflammatory profile [20]. Inspired by these anecdotes, we synthesized isonicotinates of lipophilic nature and studied their anti-inflammatory potential. Herein, we report exceptional anti-inflammatory activities of several isonicotinic acid derivatives.

## 2. Results and Discussion

To discover novel scaffolds, our molecular design was based on the documented activity of isonicotinic acid as a pharmacophoric moiety. In this context, isonicotinates were considered interesting target molecules, since isonicotinamide derivatives are mostly associated with antimicrobial activities in the literature. Isonicotinates were synthesized by employing meta- and para-linkers; whereas alkyl chains of varying lengths were used to augment or abridge the lipophilic behavior of these compounds (Figure 2).

To synthesize the target compounds, isonicotinic acid was initially reacted with meta- or para-nitro phenol using *N,N’*-dicyclohexyl carbodiimide (DCC) and 4-dimethylaminopyridine (DMAP) in dimethylformamide (DMF), which afforded **1** and **2** with 36% and 31% yields, respectively (Scheme 1). When these products were subjected to catalytic hydrogenation (Pd/C, 1–6 bar H_2_) to reduce the nitro group, hydrolysis of the reduced product returned isonicotinic acid and the corresponding aminophenol. This was presumably due to the presence of strong electron withdrawing effect of nitro group on the linker, which renders the carbon-oxygen bond in **1** and **2** considerably weak and labile under the catalytic hydrogenation conditions. An alternative strategy was adopted involving protection–deprotection method (Scheme 2). An equimolar mixture of meta- or para-substituted aminophenol was treated with di-tert-butyl pyrocarbonate to afford amine protected intermediates. This was confirmed by the presence of a signal (δ ~1.51 ppm) due to C(CH_3_)_3_ group in the ^1^H-NMR spectrum. The *N*-Boc-protected aminophenols were then reacted with isonicotinic acid using DCC/DMAP in DMF to afford **3** and **4** in satisfactory yields (41% and 36%, respectively).

These intermediates were then treated with trifluoroacetic acid (TFA) and dichloromethane (1:1) mixture at 0 °C to get free amines **5** and **6**, which were then reacted with equimolar mixture of different acid anhydrides of varying alkyl chains to afford lipophilic isonicotinate derivatives **7a**–**7e** and **8a**–**8e** in appreciable yields, thus rendering this protocol practical and straightforward to implement. The structures of the synthesized compounds were confirmed by various spectroscopic techniques. Spectral data of all the active compounds is presented in the Appendix A.

The amassed collection of compounds was assayed for anti-inflammatory activity. The oxidative burst assay was performed using chemiluminescence technique to evaluate their anti-inflammatory potential [21]. Briefly, the compounds were incubated with human blood (25 µL) diluted with Hank’s Balanced Salt Solution (HBSS) in 96-well half-area plates in the thermostatic chamber of luminometer, and serum-opsonized zymosan (SOZ) and luminol were then added. The inhibitory effect of these compounds on the production of reactive oxygen species by human blood cells was measured, and their half-maximal inhibitory concentration (IC_50_) values are presented in Table 1. The in vitro results demonstrate that both compounds **5** and **6**, which are devoid of lipophilic chain, exhibit high anti-inflammatory activity. Compound **5** is an isonicotinate of meta-aminophenol and is the most potent compound among all the synthesized compounds. An IC_50_ value of 1.42 ± 0.1 µg/mL for **5** was eight-fold better than that of standard drug Ibuprofen (IC_50_ = 11.2 ± 1.9 µg/mL). When a lipophilic acyl chain is introduced into the nitrogen atom of the aminophenol linker, a significant change in their potency was observed. For compounds with a meta-aminophenol type linker (**7a**–**7e**), only a weak activity was observed. On the other hand, for compounds with a para-aminophenol linker (**8a**–**8e**), a wider range of anti-inflammatory potency was observed. Compound **8b** with a butyryl group has an IC_50_ of 3.7 ± 1.7 μg/mL, which is three-fold better than Ibuprofen, while compound 8d with an octanoyl group is devoid of any activity. It is interesting to note that compounds **6** and **8a** (with an acetyl group) show comparable values of % inhibition at a dose of 25 µg/mL; however, **6** has a much lower IC_50_ value (8.6 ± 0.5 µg/mL) than **8a** (19.6 ± 3.4 µg/mL). The cogent manipulation of lipophilicity in our synthesized compounds results is a positive contribution towards anti-inflammatory activity of isonicotinates containing a para-aminophenol linker. Lipophilic chains of moderate length were found more effective as compared to the shorter or longer chains. For example, compound **8b** having a butyryl group had five-fold better anti-inflammatory activity than **8a** with an acetyl group. These results overall demonstrate the indispensable significance of suitably positioned linkers in isonicotinates.

In order to gain insight into the binding mode of the potent isonicotinates and look for clues to assist in the lead-to-candidate optimization, a docking study using Molecular Operating Environment (MOE) version 2019.01 builder module that allows examining a potential binding mode to the enzyme was carried out. Heteroatom number designation (Figure 3) in isonicotinates was generated by a webserver, as shown below:

Reactive oxygen species (ROS) or free radicals is a prominent factor of producing the inflammation. Cyclooxygenase-2 (COX-2) inhibition by nonsteroidal anti-inflammatory drugs (NSAIDs), like Ibuprofen, can relieve the symptoms of inflammation and pain. We therefore assume that the anti-inflammatory activity of these isonicotinoyl compounds may be correlated to their inhibitory effect on COX-2 enzyme. Hence, in silico molecular docking of isonicotinic acid derivatives was conducted on the COX-2 enzyme cavity to decipher their binding potential. Benchmarking of the docking software was done with the cognate ligand Ibuprofen (Figure 4), and docking complex images of the enzyme with compounds **5**, **6**, **8a**, and **8b** are shown in Figure 5.

The overall binding affinities of the active compounds, **5**–**6** and **8a**–**8b**, are in the range of −7.26 to −6.56 (see Table 2). These values signify strong-to-moderate interactions.

Binding energy value for Ibuprofen is −7.61. Comprehensive visual inspection of each active molecule and co-crystal ligand outlines the interaction pattern with the essential residues in the cavity. Key interactions for all compounds (**5**, **6**, **8a** and **8b**) in the binding pocket of the enzyme are listed in Table 2. For comparative analysis, the following observations were noted to interpret the active nature of isonicotinates: (i) The binding orientation (dock poses) is focused on isonicotinoyl moiety (see Figure 5); (ii) Backbone residues with prominent interaction are Val350, Leu353, Tyr356, Leu360, Tyr386, Trp388, Val524, Ala528 (hydrophobic residues), whereas Arg121 and Tyr356 are involved in hydrogen bonding, just like in Ibuprofen (Figure 4); (iii) Crucial residues are involved in different types of interaction, such as π-π and hydrogen bonding; (iv) Addition of an acyl group on the amino group of the linker decreases the electron density on the pyridine nitrogen, which in turn may reduce its capacity as a hydrogen bond acceptor and affect the orientation of the molecule in the binding pocket. For example, the pyridine nitrogen in **6** forms hydrogen bonding with Arg121 and Tyr356. In contrast, the pyridine residue in **8a** (having an acetyl group) is located further away from the Arg121 and Tyr356, while the pyridine nitrogen in **8b** (having a butyryl group) is hydrogen-bonded only to the inner NH of Arg121. Briefly, the docking study confirms that all active compounds show good binding affinity with COX-2 enzyme and are potentially good inhibitors of the enzyme, which correlates well with their potent anti-inflammatory activity demonstrated by the in vitro oxidative burst assay.

## 3. Conclusions

In conclusion, isonicotinates were synthesized by adopting efficient synthetic strategies. Isonicotinates with the least lipophilicity demonstrate exceptionally high anti-inflammatory/ROS inhibitory activities among the series. The obtained data of the anti-inflammatory activity indicate that compound **5** is a valuable candidate for further studies. Further studies are in progress to show the anti-inflammatory potential of these compounds in in vivo models.

## Data Availability

Data is contained within the article or the Appendix A.

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
