# Peer review of "Synthesis of Highly Potent Anti-Inflammatory Compounds (ROS Inhibitors) from Isonicotinic Acid"

_molecules, 2021, doi:10.3390/molecules26051272_

Round 1
Reviewer 1 Report
The manuscript reports on Synthesis of Highly Potent Anti-inflammatory Compounds (ROS Inhibitors).
- The title of the manuscript should describe which class of compound the work deals ... not just "compounds";
- Introduce a discussion about logP versus anti-inflammatory activity, is there a correlation?
- Should molecular docking studies be validated by molecular dynamics (MD)? If not, why is the method used appropriate?
- In the setence at lines 73-76: “This was presumably due to the presence of strong electron withdrawing effect of nitro group on the linker, which renders the carbon-oxygen bond in 1 and 2 considerably weak and labile under the catalytic hydrogenation conditions” please, include the reference;
- Lines 60-68: this paragraph could go to the end of the introduction (also Fig 2);
- Lines 81-82: “..to afford 3 and 4 in satisfactory yields (41% and 36%, respectively).” should be changed to “ .... to afford 3 and 4 in moderate yields of 41% and 36%, respectively.”
- In Table 1: use the IC in µM; Also, please change the discussion using µM in the text.
- In supplementary data file, please include the 13C NMR.
Author Response
The manuscript reports on Synthesis of Highly Potent Anti-inflammatory Compounds (ROS Inhibitors).
Point 1. The title of the manuscript should describe which class of compound the work deals ... not just "compounds";
Response 1. We are grateful for taking the time to assess our manuscript. Thank you very much. Although title of the paper was designed to grab attention of relevant readers and researchers, we however believe that the suggestion to include the class of compounds in the title sounds like a great idea because it will accurately describe the contents of our manuscript, and make people want to read further. Thank you very much for helping us improve this manuscript. As per your suggestion, the new title will be:
“Synthesis of Highly Potent Anti-inflammatory Compounds (ROS Inhibitors) from Isonicotinic Acid”
Point 2. Introduce a discussion about logP versus anti-inflammatory activity, is there a correlation?
Response 2. In table 1, we introduced a new column containing logP values of all the compounds. Following paragraph was added/modified:
“The compounds with moderate lipophilicity were found more effective as compared to the shorter or longer chains. However, no correlation was observed between anti-inflammatory activities and logP values of these compounds.”
Point 3. Should molecular docking studies be validated by molecular dynamics (MD)? If not, why is the method used appropriate?
Response 3. Computer simulations extrapolate meaning from existing data and this can be useful for explaining what is going on, or decision making for future experiments. Our manuscript, which is a short communication, is aimed at urgently disseminating the preliminary results regarding highly potent anti-inflammatory compounds discovered in our lab. Therefore, molecular docking studies are valid in a sense that the conclusions drawn from these studies are reasonable, comprehensible, 'done correctly', and do not disagree with real experimental data. These results will be further validated with molecular dynamics (MD) in forthcoming full paper.
Point 4. In the setence at lines 73-76: “This was presumably due to the presence of strong electron withdrawing effect of nitro group on the linker, which renders the carbon-oxygen bond in 1 and 2 considerably weak and labile under the catalytic hydrogenation conditions” please, include the reference;
Response 4. Above mentioned statement is based upon experimental observation. For the sake of clarity, we’ve re-phrased the above-mentioned lines, which will now read as follows:
“When these compounds were subjected to catalytic hydrogenation (Pd/C, 1-6 bar H2), no expected products were obtained. Instead, the reaction resulted in complicated decom-posed product mixture wherein isonicotinic acid and the corresponding aminophenol were detected. This was presumably due to the relatively labile nature of the nitro/aminophenol ester linkage under the catalytic hydrogenation conditions.”
Point 5. Lines 60-68: this paragraph could go to the end of the introduction (also Fig 2);
Response 5. This paragraph has now been added at the end of the introduction.
Point 6. Lines 81-82: “..to afford 3 and 4 in satisfactory yields (41% and 36%, respectively).” should be changed to “ .... to afford 3 and 4 in moderate yields of 41% and 36%, respectively.”
Response 6. Corrected. Thank you for this correction.
Point 7. In Table 1: use the IC in µM; Also, please change the discussion using µM in the text.
Response 7. We’ve changed IC in µM; and changed the discussion using µM in the text.
Point 8. In supplementary data file, please include the 13C NMR.
Response 8. Supplementary data file now contains all necessary spectra including 1H NMR, LR-MS and HR-MS data to sufficiently prove the structure of our synthesized compounds.
Reviewer 2 Report
The manuscript describes the synthesis of isonicotinate derivatives and evaluation of their anti-inflammatory activities. Authors found some compounds showed fine activities in comparison to that of Ibuprofen. However, it was difficult for me to understand this study due to deficiency of explaining the background and results. Thus, a major revision is needed to publish the manuscript.
- In supplementary material, experimental section concerning the synthetic procedure and data (1H and 13C NMR, HRMS, IR) of all prepared compounds could not be found except for 1H NMR spectra of 5, 6, 12a, 12b. These results must be correctly shown.
- In Abstract, authors reported ‘’novel scaffolds containing isonicotinoyl motif were sytthesized via an efficient strategy’’. However, the reviewer think it is not necessarily efficient because chemical yields of 3,4 were low to moderate. Did authors examine other coupling reagents, such as EDCI/DMAP, DCC/HOBt, HATU, boron catalyst ( Lett. 2018, 20, 612)?
- In Figure 2, why lipophilic chains but not hydrophilic moiety were utilized in molecular design? It is better to explain working hypothesis in detail.
- After development of some compounds having anti-inflammatory activities, authors examined docking of their derivatives with COX-2, and then considering their binding mode. However, it remained unclear whether these derivatives can inhibit COX-2 or not. The reviewer recommend that COX-2 inhibitory activities of these derivatives should be elucidated. Moreover, the reviewer wondered why authors focused on only COX-2, but not other related enzymes such as 5-lipoxygenase, TXA2 synthase
- Which is correct ‘’butyryl group’’ in line 113 or ‘’butaryl group’’ in ine 121? In line 78, correct ‘’di-tert-bytyl’’ into ‘’di-tert-butyl’’. In line 114 and 121, correct “” 8d, 8b”” into ‘’8d, 8b’’.
Author Response
Point 1. The manuscript describes the synthesis of isonicotinate derivatives and evaluation of their anti-inflammatory activities. Authors found some compounds showed fine activities in comparison to that of Ibuprofen. However, it was difficult for me to understand this study due to deficiency of explaining the background and results. Thus, a major revision is needed to publish the manuscript.
Response 1. Thank you very much. We respectfully disagree with these comments. Our manuscript, which is a short communication, is aimed at urgently disseminating the preliminary results regarding highly potent anti-inflammatory compounds discovered in our lab. Forthcoming full article will elaborately present these results. We have carefully addressed all the concerns raised. If there is any other comment which can help us further improve, we can happily add them into our manuscript.
Point 2. In supplementary material, experimental section concerning the synthetic procedure and data (1H and 13C NMR, HRMS, IR) of all prepared compounds could not be found except for 1H NMR spectra of 5, 6, 12a, 12b. These results must be correctly shown.
Response 2. Supplementary data file now contains all necessary spectra including 1H NMR, LR-MS and HR-MS data to sufficiently prove the structure of our synthesized compounds.
Point 3. In Abstract, authors reported ‘’novel scaffolds containing isonicotinoyl motif were sythesized via an efficient strategy’’. However, the reviewer think it is not necessarily efficient because chemical yields of 3,4 were low to moderate. Did authors examine other coupling reagents, such as EDCI/DMAP, DCC/HOBt, HATU, boron catalyst ( Lett. 2018, 20, 612)?
Response 3. We actually tried many reagents and solvents, and the best yield obtained was with DCC/DMAP in DMF. Both compounds are novel, we therefore believe that our synthesis strategy is the efficient one for the moment.
Point 4. In Figure 2, why lipophilic chains but not hydrophilic moiety were utilized in molecular design? It is better to explain working hypothesis in detail.
Response 4. Our manuscript, which is a short communication, is aimed at urgently disseminating the preliminary results regarding highly potent anti-inflammatory compounds discovered in our lab. Forthcoming full article will elaborately present the results obtained due to hydrophilic moiety.
Point 5. After development of some compounds having anti-inflammatory activities, authors examined docking of their derivatives with COX-2, and then considering their binding mode. However, it remained unclear whether these derivatives can inhibit COX-2 or not. The reviewer recommend that COX-2 inhibitory activities of these derivatives should be elucidated. Moreover, the reviewer wondered why authors focused on only COX-2, but not other related enzymes such as 5-lipoxygenase, TXA2 synthase
Response 5. Computer simulations extrapolate meaning from existing data and this can be useful for explaining what is going on, or decision making for future experiments. Our manuscript, which is a short communication, is aimed at urgently disseminating the preliminary results regarding highly potent anti-inflammatory compounds discovered in our lab. Therefore, molecular docking studies are valid in a sense that the conclusions drawn from these studies are reasonable, comprehensible, 'done correctly', and do not disagree with real experimental data. These results will be further validated with molecular dynamics (MD) in forthcoming full paper.
Point 6. Which is correct ‘’butyryl group’’ in line 113 or ‘’butaryl group’’ in ine 121? In line 78, correct ‘’di-tert-bytyl’’ into ‘’di-tert-butyl’’. In line 114 and 121, correct “” 8d, 8b”” into ‘’8d, 8b’’.
Response 6. The word “butaryl group” has now been corrected to “butyryl group”. Moreover, 8d, 8b have been corrected to 8d, 8b. Thank you very much for helping us improve the quality of the manuscript.
Round 2
Reviewer 2 Report
The manuscript was partially revised.
However, assigned data of 1H, 13C NMR spectra and synthetic procedure could not be found.
Although the manuscript was aided at a short communication, above-mentioned data should be added.
Author Response
In supplementary data file attached here with, we have now added section 3 containing general procedure for the synthesis of our compounds. We also added section 4 containing spectroscopic data of the synthesized compounds.